# SPaR: Self-Play with Tree-Search Refinement to Improve Instruction-Following in Large Language Models

**Jiale Cheng**[1,2*], **Xiao Liu**[2,3*], **Cunxiang Wang**[2,3] , **Xiaotao Gu**[2] , **Yida Lu**[1,2†], **Dan Zhang**[3] ,
**Yuxiao Dong**[3] , **Jie Tang**[3] , **Hongning Wang**[1] , **Minlie Huang**[1‡]
[1]The Conversational Artificial Intelligence (CoAI) Group, Tsinghua University
[2]Zhipu AI
[3]The Knowledge Engineering Group (KEG), Tsinghua University
 `chengjl23@mails.tsinghua.edu.cn, aihuang@tsinghua.edu.cn`

## Abstract

Instruction-following is a fundamental capability of language models, requiring the model to recognize even the most subtle requirements in the instructions and accurately reflect them in its output. Such an ability is well-suited for and often optimized by preference learning. However, existing methods often directly sample multiple independent responses from the model when creating preference pairs. Such practice can introduce content variations irrelevant to whether the instruction is precisely followed (e.g., different expressions about the same semantic), interfering with the goal of teaching models to recognize the key differences that lead to improved instruction following. In light of this, we introduce SPaR, a self-play framework integrating tree-search self-refinement to yield valid and comparable preference pairs free from distractions. By playing against itself, an LLM employs a tree-search strategy to refine its previous responses with respect to the instruction while minimizing unnecessary variations. Our experiments show that a LLaMA3-8B model, trained over three iterations guided by SPaR, surpasses GPT-4-Turbo on the IFEval benchmark without losing general capabilities. Furthermore, SPaR demonstrates promising scalability, greatly enhancing models like GLM-4-9B and LLaMA3-70B. We also identify how inference scaling in tree search would impact model performance. Our code and data are publicly available at `https://github.com/thu-coai/SPaR`.

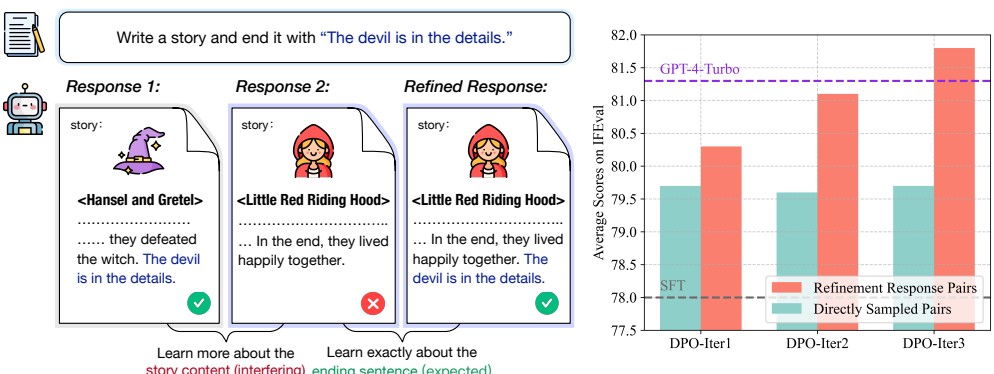

Figure 1: An example of the interfering factors (*story content*) in independently sampled multiple responses (Left). Refined response pairs exclude these factors, highlight the key difference (*ending sentence*), and lead to improved performance on iteratively trained LLaMA3-8B-Instruct (Right).

---

[*] Equal contributions.

[†]Work done when JC and YL interned at Zhipu AI.

[‡]Corresponding author

# 1 INTRODUCTION

To date, Large Language Models (LLMs) have achieved great success in a wide range of tasks (Brown et al., 2020; Zeng et al., 2022; Chowdhery et al., 2023; Touvron et al., 2023; GLM et al., 2024). As LLMs are applied to various scenarios, their instruction-following capability becomes crucial (Ouyang et al., 2022; Bai et al., 2022), especially to follow instructions with multiple constraints (Zeng et al., 2023; Zhou et al., 2023; Jiang et al., 2023b). The failure to accurately follow instructions can even lead to safety issues (Ruan et al., 2023).

Subtle nuances can determine the success of instruction-following tasks (Zhou et al., 2023), making preference learning (Rafailov et al., 2024; Hou et al., 2024) a well-suited solution. However, existing methods usually sample multiple independent responses from the target model (Yuan et al., 2024; Wu et al., 2024; Dong et al., 2024), inadvertently introducing irrelevant variations to whether the instruction was successfully followed. As illustrated in Figure 1, given the instruction: "Write a story and end it with *The devil is in the details*", sampling multiple independent responses from an LLM can result in responses as different as the story *Little Red Riding Hood* vs. *Hansel and Gretel*. This variation in the narrative content can interfere with the model's ability to learn how to realize the critical requirement—the specified ending sentence—and ultimately mislead the comparison within the preference pair. Therefore, effective learning from preference pairs necessitates excluding these extraneous factors and focusing on the key differences that drive the success of instruction-following.

In this paper, we propose SPAR, a self-play method integrated with tree-search refinement to enhance instruction-following capabilities of LLMs. The key lies in iteratively teaching LLMs to learn instruction-following from nuances by playing against itself with structured tree search. In each turn of self-play, an LLM takes two different roles: the actor and the refiner, which are both initialized from the same model. The actor executes complex instructions while the refiner critiques and refines the actor's responses. During the iteration, we first collect the actor's responses which fail to follow the instructions accurately, as judged by the refiner. Starting from those failed responses, we apply a tree-search algorithm for refinement, which ensures consistent improvements against previous turns and naturally creates valid comparison counterparts for model training.

We conduct experiments on several LLMs, LLaMA3 series (MetaAI, 2024), GLM-4-9B (GLM et al., 2024), and Mistral-7B-Instruct (Jiang et al., 2023a), over multiple iterations. Through extensive experiments, we demonstrate significant improvements in the models' instruction-following capability, outperforming other self-improvement methods such as self-rewarding (Yuan et al., 2024) and meta-rewarding (Wu et al., 2024). Notably, after three iterations, SPAR improves LLaMA3-8B-Instruct over GPT-4-Turbo on the IFEval benchmark (Zhou et al., 2023). Moreover, scaling test-time compute by integrating tree-search refinement during inference can further improve the quality of instruction following. Additionally, we find that with several iterations, the refiner's judgment and refinement capabilities can match or even exceed those of the distilled LLM, indicating great potential for continuous self-improvement without being limited by the initial bootstrapping data. Ablation studies demonstrate the importance of each component within our framework. Importantly, our method does not degrade performance on general benchmarks. In summary, our contributions are:

- We reveal that preference pairs derived from independently sampled responses often contain interfering factors, hampering preference learning to improve instruction following. As a result, a performing solution has to minimize such interference and highlight the key differences contributing to the success of instruction following.
- We introduce SPAR, a novel self-play framework that enables continuous self-improvement in instruction-following tasks. Through three iterations, our method boosts LLaMA3-8B-Instruct to achieve GPT4-level performance and scales effectively to enhance LLaMA3-70B-Instruct.
- We construct a high-quality dataset with 43K complex instruction-following prompts and an SFT dataset that can improve the instruction-following capabilities of LLMs.

# 2 METHOD

We introduce SPAR, an automated and scalable approach designed for self-improvement of instruction-following tasks through self-play. The core idea is to create paired responses with

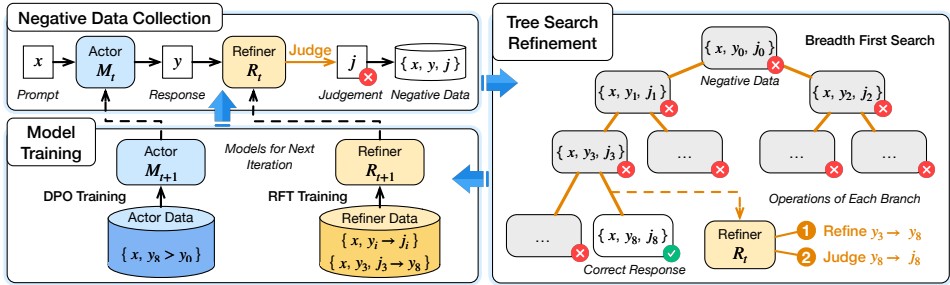

Figure 2: SPaR iterative training framework. At iteration $t$, the refiner $R_t$ first judges the generated responses from the actor $M_t$ to collect negative data. Next, a tree-search algorithm is employed to refine these imperfect responses. Finally, using the data from the above steps, we can optimize the actor and refiner for the next iteration, aiming for continuous self-improvement.

minimal irrelevant variations, thereby highlighting the key differences that manifest the success of instruction-following.

## 2.1 OVERALL FRAMEWORK

The overall framework of SPaR is illustrated in Figure 2. Briefly, our framework involves an actor model and a refiner model, which are both initialized from the same base model. The actor generates responses to given instructions while the refiner judges and refines these responses. This iterative self-play process, involving response generation, judgment, and refinement, fosters continuous self-improvement.

Formally, in each iteration, given an instruction $x$ from the prompt set, the actor generates a response $y$. The refiner identifies the responses that do not follow the instructions accurately, termed as negative responses. Our objective is to refine the negative response (represented as $y_0$ in Figure 2) into a correct response (represented as $y_8$ in the figure). These generated refinement pairs, e.g., $(x, y_8 > y_0)$, are collected and used to optimize the actor via Direct Preference Optimization (DPO) (Rafailov et al., 2024). Simultaneously, we apply Rejection-sampling Fine-Tuning (RFT) (Yuan et al., 2023) to improve the refiner. This process prepares both models for the next iteration of self-improvement.

In this iterative process, we face two major challenges: the scarcity of complex instruction-following data and the difficulty of achieving successful refinements. To address the lack of high-quality, multi-constraint instruction-following datasets, we generate complex instructions using a taxonomy-based approach and create corresponding SFT datasets to initialize the actor and refiner models (§2.2). To ensure a high success rate in refining negative responses, we employ a tree search strategy that systematically explores refinement paths and facilitates subsequent training (§2.3).

## 2.2 DATA CONSTRUCTION

### 2.2.1 PROMPT CREATION

Given the scarcity of high-quality data for instruction-following tasks, especially those with multiple constraints, we start by creating a high-quality dataset of instruction-following prompts.

**Seed Prompts.** To ensure the quality and diversity of our dataset, and to prevent issues like insufficient diversity or even model collapse (Liu et al., 2024; Shumailov et al., 2024), we use a seed set of prompts derived from the Infinity-Instruct dataset (Zhao et al., 2024), which contains ten million high-quality conversations. After applying rule-based filtering based on length, keywords, and self-BLEU, we obtain approximately 50k seed prompts.

**Taxonomy-based Prompt Construction.** Complex prompts constructed without human intervention tend to be poorly diversified, as the types of constraints added are often distributed unevenly (Sun et al., 2024). Therefore, we adopt a taxonomy-based mechanism to make constraint types comprehensive and balanced. Our taxonomy for instruction-following constraints is derived from Cheng et al. (2024) and further refined to be more comprehensive. After building the constraint taxonomy, we employ it to construct complex instruction-following tasks based on seed prompts. We sample a

main constraint type and employ a strong LLM to add several other constraints to make the original prompt more complex. Moreover, we leverage the strong LLM to assess the validity of the generated prompt, ensuring that the constraints do not conflict with each other or create unreasonable scenarios with the original task. The detailed taxonomy and prompt can be found in Appendix A.

### 2.2.2 ACTOR AND REFINER INITIALIZATION

The taxonomy-based prompt construction results in about 43k prompts. We utilize 8k prompts for actor initialization, another 5k for the refiner, and save 30k for further self-play training.

**Actor Data Creation.**   To bootstrap the actor model with strong instruction-following capabilities, we first collect a strong LLM's responses to these complex prompts, thereby producing supervised fine-tuning (SFT) data $(x, y) \in D_{\text{Actor}}$ for the actor model, where $x$ is the complex instruction and $y$ is the strong LLM's response. Then, we fine-tune the base model to get an initial actor $M_0$.

**Refiner Data Creation.**   To bootstrap the refiner model with strong judgment and refinement capability, we sample responses from the initial actor $M_0$. Then, we collect the judgments from a strong LLM to form a dataset, $(x, y, j) \in D_{\text{JSFT}}$. We collect responses that are judged not to accurately follow instructions and term them as negative responses. For these negative responses, we use the strong LLM to correct them with minimal revisions to avoid irrelevant variations. In this way, we get a refinement dataset, $(x, y_{\text{negative}}, j, y_{\text{refined}}) \in D_{\text{RSFT}}$. The refiner is then trained with $D_{\text{Refiner}} = D_{\text{JSFT}} \cup D_{\text{RSFT}}$ to create the initial refiner $R_0$.

**Training Strategy.**   For both actor and refiner models, we use standard supervised fine-tuning with the loss function:

$$\mathcal{L} = -\frac{1}{N} \sum_{i=1}^{N} \log P(r_i | q, r_{<i}), \tag{1}$$

where $q$ denotes the input, $r$ signifies the target response, and $N$ represents the length of $r$. For actor training, we have input $q = x$ and target $r = y$. When it comes to the refiner, we use input $q = (x, y)$ and target $r = j$ for $D_{\text{JSFT}}$, and input $q = (x, y_{\text{negative}}, j)$ and target $r = y_{\text{refined}}$ for $D_{\text{RSFT}}$.

### 2.3 TREE-SEARCH INTEGRATED SELF-PLAY TRAINING

After initializing the actor and refiner models, we embark on an iterative process for continuous self-improvement. In each iteration, we first collect the negative data, where the responses fail to accurately follow the instructions (§2.3.1). Then, we utilize a tree-search algorithm to refine the negative responses (§2.3.2) and form the training data for the next iteration of the actor (§2.3.3) and refiner (§2.3.4). This iterative self-play pipeline allows us to continuously improve both models.

### 2.3.1 NEGATIVE DATA COLLECTION

For each prompt $x$, we first sample $K$ responses $\{y_1, y_2, \ldots, y_K\}$ from the actor model. This step ensures that there are enough negative responses to support subsequent learning. Then, for each prompt and response pair, we utilize the refiner to generate a judgment, which contains two parts: a label suggesting whether the response follows the instruction and an explanation about the assessment. To make this judgment more accurate, we incorporate the self-consistency mechanism (Wang et al., 2022), which is also applied in the subsequent refinement process. Specifically, we obtain multiple judgments from the refiner and determine the final label through majority voting, as detailed in Appendix E.4. After majority voting, we randomly select one judgment that matches the voted label to serve as the final judgment. This process allows us to identify challenging prompts that elicit responses that do not accurately follow the instructions, yielding tuples in the form of $(x, y_{\text{negative}}, j)$, where $y_{\text{negative}}$ is the incorrect response and $j$ is its corresponding judgment.

### 2.3.2 TREE-SEARCH REFINEMENT

After collecting these negative instances, the core step is to refine the responses to form preference pairs. These self-refined pairs are crucial for highlighting the subtle differences that can determine the success of instruction-following tasks, thereby facilitating effective learning. Given that direct

refinement often results in a low success rate, we employ a tree-search approach. We implement both breadth-first search (BFS) and depth-first search (DFS) strategies for this refinement. Detailed algorithms for these methods are provided in Appendix B.

To illustrate our process, we take BFS as an example and illustrate the procedure in Figure 2. Starting with an incorrect instruction-response pair and its judgment as the root node, we expand the search tree level-by-level until a correct response is found. At each intermediate node, we generate potential refinements for the current response and evaluate its correctness using the refiner. The number of generated refinements corresponds to the number of branches. Specifically, at a level of the tree, the refiner: 1). generates potential refinements for each node in the current level; 2). judges the correctness of these refinements. This creates a set of child nodes with new responses and their corresponding judgments. The search process continues until we obtain a tuple $(x, y_{\text{negative}}, y_{\text{refined}})$, where $y_{\text{refined}}$ is the newly refined, correct response. Importantly, SPAR combines the strengths of both tree-search and self-refinement, exploring multiple refinement paths while minimizing the interfering factors, producing effective preference learning data.

### 2.3.3 ACTOR TRAINING

To optimize the actor model, we leverage the refinement pairs for preference learning using DPO. At iteration $t$, we train the actor model $M_t$ with refinement pairs $(y_{\text{negative}}, y_{\text{refined}})$, treating $y_{\text{negative}}$ as the rejected response $(y_l)$ and $y_{\text{refined}}$ as the chosen response $(y_w)$. The training dataset is denoted as $D_{\text{dpo}}^t$ and the DPO loss is described as follows:

$$\mathcal{L}_{\text{DPO}}(\pi_\theta^t; \pi_{\text{ref}}) = -\mathbb{E}_{(x, y_w, y_l) \sim D_{\text{dpo}}^t} \left[ \log \sigma \left( \beta \log \frac{\pi_\theta^t(y_w|x)}{\pi_{\text{ref}}(y_w|x)} - \beta \log \frac{\pi_\theta^t(y_l|x)}{\pi_{\text{ref}}(y_l|x)} \right) \right] \qquad (2)$$

where $\pi_\theta^t$ represents the actor model $M_t$, and the reference model $\pi_{ref}$ initialized with $M_t$ remains fixed during the training process. This results in a new actor model, $M_{t+1}$, for the next iteration.

### 2.3.4 REFINER TRAINING

Given that the input for the refiner is templated, we use RFT to obtain the new refiner $R_{t+1}$. The RFT training data consists of two components: the refinement data and the judgment data for improving the refiner's corresponding capabilities.

**Refinement Training Data.** The refinement training data consists of tuples that capture the process of refining incorrect responses. For each incorrect response from the tree-search based refinement step, we collect tuples in the form of $(x, y_p, j_p, y_{\text{refined}})$, where $(x, y_p, j_p)$ represents the parent node of the final correct response in the refinement tree, and $y_{\text{refined}}$ is the correctly refined response.

**Judgment Training Data.** The judgment training data is derived both from the negative data collection and nodes of the tree-search process. This dataset consists of tuples $(x, y_i, j_i)$, where $x$ is the prompt, $y_i$ is a response to $x$, and $j_i$ is the judgment consistent with majority voting.

Then, we perform supervised fine-tuning using the constructed training data. For the refinement data $D_{\text{refine}}^t$ we use the tuples $(x, y_p, j_p, y_{\text{refined}})$ with input $q = (x, y_p, j_p)$ and target $r = y_{\text{refined}}$. For the judgment data $D_{\text{judge}}^t$, we use the tuples $(x, y_i, j_i)$ with input $q = (x, y_i)$ and target $r = j_i$. The supervised fine-tuning loss is given by Eq (1). By employing this self-play training process with the tree-search based self-refinement strategy, SPAR iteratively enhances both the actor and refiner models, aiming for continuous self-improvement in instruction-following tasks.

## 3 EXPERIMENTS

### 3.1 EXPERIMENT SETUP

**Backbone Models.** We have conducted experiments on several popular LLMs:

- **LLaMA3 Series** (MetaAI, 2024) are the best-performing models of their size, showcasing top-tier instruction-following capabilities among open-source LLMs.

- **GLM-4-9B-Chat** (GLM et al., 2024) excels in instruction-following tasks, offering competitive performance under 10B parameters.
- **Mistral-7B-Instruct** (Jiang et al., 2023a) is one of the most popular LLMs and has shown good performance across a wide range of tasks.

**Settings.** In this work, we focus on enhancing the instruction-following abilities of LLMs in a self-play fashion. We utilize SFT to bootstrap models under 10B parameters as actor and refiner models. For the more advanced LLaMA3-70B-Instruct, we directly employ it in both roles. Following this, we perform a three-iteration self-play training using 10k prompts per iteration from our generated dataset. In each iteration, we apply DPO for the actor and RFT for the refiner. We refer to the trained LLaMA3-8B-Instruct as SPAR-8B, LLaMA3-70B-Instruct as SPAR-70B, GLM-4-9B-Chat as SPAR-9B, and Mistral-7B-Instruct as SPAR-7B. More implementation details can be found in Appendix C. Description of baseline methods is provided in Appendix D.

## 3.2 EVALUATION BENCHMARKS

As both the actor and refiner continually evolve within our framework, it's crucial to comprehensively evaluate both of their capabilities.

**Actor's Instruction-following Capability.** To assess the actor's ability to follow instructions, we rely on two widely-used benchmarks: IFEval (Zhou et al., 2023) and FollowBench (Jiang et al., 2023b). IFEval offers 541 verifiable instructions specifically designed for code-based evaluation. These instructions cover 25 verifiable types, including tasks like *Keyword Frequency* and *Number of Words*. FollowBench, on the other hand, encompasses five categories of more subjective constraints: *Content*, *Situation*, *Style*, *Format*, and *Example*. This dataset features 820 meticulously curated instructions across five difficulty levels and utilizes a hybrid assessment approach combining rule-based and LLM-as-judge evaluations.

**Refiner's Judgment and Refinement Capability.** For assessing the refiner's judgment capability, we turn to LLMBar (Zeng et al., 2023), a dataset designed to measure the assessment ability of LLMs in the context of instruction-following tasks. LLMBar includes 419 instruction-response pairs, categorized into two subsets: *Natural* and *Adversarial*. Originally, the task involves pair-wise comparisons to identify successful and failed responses. We adapted it to a point-wise judgment task, asking the model to determine whether each instruction-following task is successful.

To evaluate the refiner's capability in refinement, we split 200 samples from the $D_{\text{RSFT}}$ to create a test set, and we employ both GPT-4o and SPAR-8B-RFT-iter3, the refiner after three rounds of training, as judges to evaluate whether the refined responses are accurately following the instructions.

## 3.3 ACTOR EVALUATION RESULTS

**SPAR significantly improves instruction-following ability.** As illustrated in Table 1, the iteratively trained LLMs demonstrate substantial improvements in both the IFEval and FollowBench benchmarks. Remarkably, after three training iterations, SPAR-8B-DPO-iter3 even surpasses GPT-4-Turbo (81.3% average accuracy) on IFEval. Moreover, incorporating the tree-search refinement technique during the inference stage significantly boosts performance. Additionally, the SPAR showcases excellent scalability with respect to model size, which substantially enhances the instruction-following abilities of the LLaMA3-70B-Instruct model.

**SPAR does not damage general abilities.** As shown in Appendix E.2, we assessed each iteration's performance on general benchmarks, including GSM8k (Cobbe et al., 2021), TriviaQA (Joshi et al., 2017), MMLU (Hendrycks et al., 2020), and HumanEval (Chen et al., 2021). The results indicate that SPAR maintains or even improves general performance, particularly on GSM8k and HumanEval benchmarks, demonstrating that enhanced instruction-following capabilities support overall LLM alignment.

**SPAR outperforms other baselines significantly.** Figure 3 demonstrates the improvements on IFEval with each training iteration. In every iteration, SPAR outperforms other methods. Notably,

Table 1: Main results of iteratively trained LLMs on instruction-following benchmarks (Cf. Table 6 for full results). P stands for prompt level, and I represents instruction level. L and S denote loose and strict evaluations, respectively. Avg. indicates average results and Lv means level. Results using inference-time tree search are highlighted in green . The highest results for each backbone model is **bolded**. Scores marked with $^{\dagger}$ are sourced directly from the original paper.

| Model | IFEval | | | | | FollowBench (SSR) | | | | | |
|---|---|---|---|---|---|---|---|---|---|---|---|
| | P (L) | I (L) | P (S) | I (S) | Avg. | Lv-1 | Lv-2 | Lv-3 | Lv-4 | Lv-5 | Avg. |
| *LLaMA3-8B Models* | | | | | | | | | | | |
| LLaMA3-8B-Instruct | 77.6 | 84.5 | 70.6 | 78.9 | 77.9 | 69.4 | 62.2 | 63.1 | 61.9 | 60.9 | 63.5 |
| AutoIF-8B$^{\dagger}$ | 43.1 | 56.0 | 28.8 | 42.2 | 42.5 | 54.6 | 52.1 | 50.0 | 49.0 | 43.7 | 49.9 |
| SELF | 78.2 | 84.5 | 76.0 | 82.9 | 80.4 | 68.3 | 65.7 | 65.2 | 62.2 | 62.4 | 64.8 |
| Humpback | 72.5 | 80.2 | 70.1 | 78.1 | 75.2 | 66.8 | 66.1 | 67.2 | 60.2 | 62.6 | 64.6 |
| Self-Rewarding | 77.3 | 84.2 | 74.1 | 81.7 | 79.3 | 72.8 | 66.6 | 66.8 | **64.9** | 64.1 | 67.0 |
| Meta-Rewarding | 77.8 | 84.1 | 75.4 | 82.3 | 79.9 | 73.9 | 71.9 | 66.0 | 62.3 | 62.6 | 67.3 |
| SPAR-8B-SFT | 75.4 | 82.5 | 73.4 | 80.6 | 78.0 | 73.9 | 67.4 | 68.1 | 63.1 | 61.3 | 66.8 |
| SPAR-8B-DPO-iter1 | 78.0 | 84.7 | 75.8 | 82.6 | 80.3 | **75.3** | 67.7 | 67.6 | 64.7 | 62.3 | 67.5 |
| SPAR-8B-DPO-iter2 | 78.9 | 85.0 | 77.1 | 83.3 | 81.1 | 73.9 | 71.9 | 69.1 | 64.0 | 62.2 | 68.2 |
| SPAR-8B-DPO-iter3 | **79.9** | **85.4** | **78.0** | **83.7** | **81.8** | 73.0 | **72.3** | **70.0** | 64.1 | **64.7** | **68.8** |
| w/ tree search | 82.4 | 87.5 | 79.5 | 85.3 | 83.7 | 73.9 | 71.7 | 70.3 | 66.8 | 64.1 | 69.4 |
| *GLM-4-9B Models* | | | | | | | | | | | |
| GLM-4-9B-Chat | 71.5 | 79.9 | 68.0 | 77.2 | 74.2 | 80.8 | 75.1 | 67.4 | 64.3 | **65.4** | 70.6 |
| SPAR-9B-SFT | 71.5 | 80.5 | 68.8 | 78.1 | 74.7 | 79.4 | 70.9 | 68.2 | 65.1 | 63.7 | 69.5 |
| SPAR-9B-DPO-iter3 | **77.3** | **84.1** | **73.6** | **81.4** | **79.1** | **82.7** | **76.7** | **67.9** | **68.3** | 64.2 | **72.0** |
| *LLaMA3-70B Models* | | | | | | | | | | | |
| LLaMA3-70B-Instruct | 83.7 | 88.9 | 77.1 | 83.8 | 83.4 | 77.1 | 72.5 | 69.4 | 68.7 | 66.3 | 70.8 |
| AutoIF-70B$^{\dagger}$ | **85.6** | **90.4** | 80.2 | 86.7 | 85.7 | 71.0 | 67.2 | 66.2 | 64.6 | 63.5 | 66.5 |
| SPAR-70B-DPO-iter3 | **85.6** | 90.2 | **81.3** | **87.3** | **86.1** | 80.3 | 75.7 | 71.4 | 73.7 | 70.5 | 74.3 |

even after three iterations, other methods fail to surpass the performance of SPAR's first iteration. Generally, our method and SELF outperform self-rewarding and meta-rewarding approaches, underscoring the importance of learning from refinement and excluding the interfering factors in instruction-following tasks. Furthermore, SPAR's superior performance compared to SELF indicates that contrastive refinement response pairs can highlight key differences, which are difficult to learn using only correct responses. Additionally, only SPAR-8B-SFT outperforms the original LLaMA3-8B-Instruct, which suggests that incorporating the judgment SFT or refinement SFT data would reduce performance, likely due to the huge task gap and reduced diversity in the data.

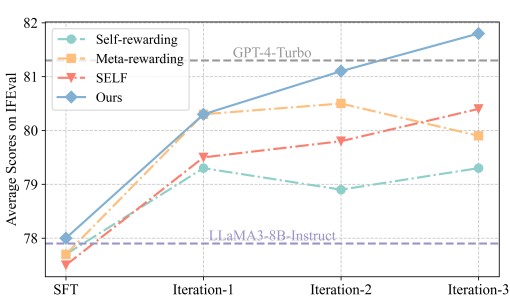

Figure 3: Comparison with baseline methods across iterations (Cf. Figure 9 for SPAR-7B). SPAR-8B consistently surpasses all baselines.

## 3.4 REFINER EVALUATION RESULTS

**SPAR iteratively enhances judgment capability.** Our analysis in Table 2 shows that SPAR iterations notably improve the model's ability to evaluate instruction-following tasks. By iteration three, the refiner SPAR-8B-RFT-iter3 surpasses GPT-4o-Mini, the model used to construct the judgment SFT dataset. This finding highlights the potential for continuous self-improvement, as the supervised fine-tuning data is not a bottleneck. Interestingly, our refiner greatly outperforms GPT-4o-Mini on adversarial test sets, suggesting that the similar positive and negative examples generated during tree search can make our model more robust against adversarial samples.

Table 2: Evaluation of judgment capability for iteratively trained LLMs on LLMBar. (Cf. Table 8 for Mistral-7B-Instruct results.) Acc. denotes accuracy. The highest scores for each base model are highlighted in **bold**.

| Model | Natural | | Adversarial | | | | | | | | | | | Average | |
| --- | --- | --- | --- | --- | --- | --- | --- | --- | --- | --- | --- | --- | --- | --- | --- |
| | | | GPTInst | | GPTOut | | Manual | | Neighbor | | Average | | | | |
| | Acc. | F1 | Acc. | F1 | Acc. | F1 | Acc. | F1 | Acc. | F1 | Acc. | F1 | | Acc. | F1 |
| GPT-4o-Mini | 74.5 | 70.5 | 69.2 | 61.6 | 60.9 | 51.4 | 59.8 | 51.9 | 72.8 | 66.4 | 65.7 | 57.8 | | 67.4 | 60.4 |
| *LLaMA3-8B Models* | | | | | | | | | | | | | | | |
| LLaMA3-8B-Instruct | 60.0 | 51.8 | 55.4 | 46.1 | 47.9 | 39.5 | 51.1 | 36.6 | 54.5 | 45.0 | 52.2 | 41.8 | | 53.8 | 43.8 |
| SELF | 69.5 | 61.6 | 62.0 | 50.7 | 64.9 | 54.8 | 57.6 | 41.8 | 64.6 | 51.3 | 62.2 | 49.6 | | 63.7 | 52.0 |
| Self-Rewarding | **71.0** | **66.3** | 70.1 | **66.7** | 63.8 | 59.5 | 62.0 | 55.7 | 67.5 | 61.7 | 65.9 | 60.9 | | 66.9 | 61.9 |
| Meta-Rewarding | 70.5 | **66.3** | 68.5 | 64.6 | 64.9 | **60.2** | 64.1 | 58.3 | **69.0** | **63.1** | 66.6 | 61.6 | | 67.4 | 62.5 |
| SPaR-8B-SFT | 68.5 | 60.9 | 67.9 | 62.4 | 59.6 | 50.0 | 63.0 | 54.1 | 68.3 | 59.3 | 64.7 | 56.5 | | 65.5 | 57.3 |
| SPaR-8B-RFT-iter1 | 68.5 | 63.2 | 66.8 | 60.6 | 63.8 | 55.3 | 62.0 | 53.3 | 66.8 | 59.0 | 64.9 | 57.1 | | 65.6 | 58.3 |
| SPaR-8B-RFT-iter2 | 70.5 | 64.2 | 66.8 | 61.6 | **66.0** | 60.0 | 65.2 | 57.9 | **69.0** | 62.4 | 66.8 | 60.5 | | 67.5 | 61.2 |
| SPaR-8B-RFT-iter3 | 70.5 | 65.9 | **70.7** | **66.7** | 63.8 | 57.5 | **68.5** | **63.3** | 68.3 | 62.2 | **67.8** | **62.4** | | **68.3** | **63.1** |
| *GLM-4-9B Models* | | | | | | | | | | | | | | | |
| GLM-4-9B-Chat | **74.5** | **76.5** | 74.5 | **75.9** | 57.4 | **62.3** | 53.3 | 56.6 | 69.8 | **72.0** | 63.7 | **66.7** | | 65.9 | **68.6** |
| SPaR-9B-SFT | 70.5 | 65.5 | 72.8 | 70.2 | **59.6** | 55.8 | 64.1 | 53.5 | 71.3 | 67.2 | 66.9 | 61.7 | | 67.7 | 62.5 |
| SPaR-9B-RFT-iter3 | 71.0 | 68.8 | **75.5** | 74.6 | 58.5 | 55.2 | **68.5** | **64.2** | 68.7 | 65.9 | **67.8** | 64.9 | | **68.4** | 65.7 |
| *LLaMA3-70B Models* | | | | | | | | | | | | | | | |
| LLaMA3-70B-Instruct | 75.0 | 71.9 | 73.4 | 69.6 | **69.1** | **66.7** | 66.3 | **60.8** | 69.0 | 63.4 | 69.5 | 65.1 | | 70.6 | 66.5 |
| SPaR-70B-RFT-iter3 | **78.0** | **74.7** | **78.8** | **76.9** | 64.9 | 61.2 | **67.4** | 59.5 | **72.4** | **68.1** | **70.9** | **66.4** | | **72.3** | **68.1** |

**SPaR progressively improves refinement capability.** Table 3 demonstrates continuous improvement in refinement accuracy (success rate) of LLaMA3-8B-Instruct with each training iteration, eventually matching the level of GPT-4o-Mini, the strong LLM for SFT data construction. This further showcases a promising way for self-evolution in instruction-following tasks. However, it also points to a potential issue of self-evaluation bias: when the refiner self-evaluates refinement accuracy, it performs significantly better than when evaluated by GPT-4o.

Table 3: Refinement evaluation results. Acc-GPT uses GPT-4o as judge; -SPaR uses SPaR-8B-RFT-iter3.

| Model | Acc-GPT | Acc-SPaR |
| --- | --- | --- |
| GPT-4o-Mini | **79.0** | 71.0 |
| SPaR-8B-SFT | 73.5 | 71.0 |
| SPaR-8B-RFT-iter1 | 77.5 | 77.0 |
| SPaR-8B-RFT-iter2 | 74.5 | 76.0 |
| SPaR-8B-RFT-iter3 | **79.0** | **90.5** |

## 3.5 Ablations and Analysis

**Refinement preference pairs enhance instruction-following capability more effectively.** To verify that the interfering factors indeed affect preference learning and motivate the need to highlight the key differences, we have conducted a synthetic data experiment featuring two tasks:

- *Character Sequence Generation*: The model needs to generate a specified number of given letters, with no restrictions on letter case, such as generating 12 letters a. For each prompt, we first construct a negative response in lowercase. In order to introduce disturbing factors, we have the correct response in uppercase for interfering pairs while maintaining refined pairs lowercase correctness.
- *Start/End Story Generation*: The model is asked to generate a story that starts with sentence 1 and ends with sentence 2. The negative response lacks either sentence 1 or 2. Interfering pairs have a different story concatenated with these sentences; refined pairs keep the same story intact.

Figure 4 shows that refinement pairs significantly outperform interfering pairs in both tasks, with larger and more effective improvements. Particularly in story generation, diverging stories results in worse accuracy than the original model. Moreover, in the character generation task, we can clearly observe that the interfering factor (uppercase ratio) is learned quickly. However, the task is not performed as well as the refinement setting, highlighting the necessity of focusing on key differences and excluding possible interfering factors.

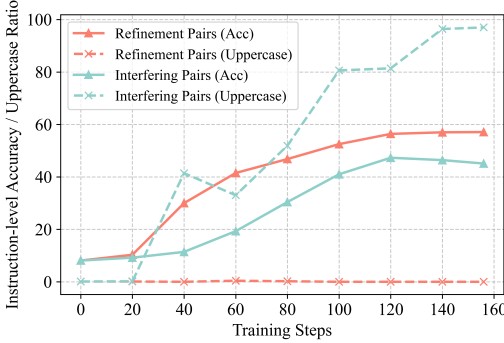 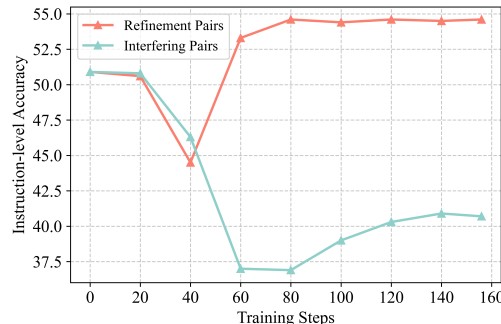

Figure 4: Synthetic data experiment results: *Character Sequence Generation* (left) and *Start/End Story Generation* (right). For *Character Sequence Generation*, interfering pairs show rapid learning of the uppercase ratio (interfering factor) but perform worse than refinement pairs. In the *Start/End Story Generation* task, refinement pairs outperform interfering pairs, which even underperform the original model at step 0.



Table 4: Ablation study on the actor.

| Model | IFEval | | FollowBench (SSR) |
|---|---|---|---|
| | Prompt(S) | Instruction(S) | Avg. |
| SPAR-8B-DPO-iter3 | 78.0 | 83.7 | 68.8 |
| *w/o* Tree Search | -2.0 | -0.8 | -1.7 |
| *w/o* Iterative Training | -0.9 | -0.2 | -2.0 |
| *w/o* Refinement | -2.6 | -1.6 | -3.1 |

Table 5: Ablation study on the refiner.

| Model | Natural | | Adversarial | |
|---|---|---|---|---|
| | Acc. | F1 | Acc. | F1 |
| SPAR-8B-RFT-iter3 | 70.5 | 65.9 | 67.8 | 62.4 |
| *w/o* Tree Search | -0.5 | -1.2 | -4.3 | -8.2 |
| *w/o* Iterative Training | -0.5 | -2.5 | -1.7 | -3.5 |



Furthermore, the ablation study on actor's performance in Table 4 further reveals a significant drop when refinement data is omitted. SPAR's superiority over self-rewarding and meta-rewarding methods in Table 1 also underscores the importance of using refinement pairs to eliminate interfering factors. Additionally, the string-level similarity of refinement response pairs is 0.90, much higher than 0.85 of the independently sampled response pairs.

**Each element is crucial in SPAR.** The primary elements of SPAR include the tree-search refinement process and iterative training. We thus conduct ablation studies to assess the significance of these elements. For the tree-search process, as shown in Table 4, excluding tree search significantly reduces the actor's performance. This might be due to a lack of difficult samples that require more iterations to refine and a reduced number of preference pairs. Table 10 illustrates that tree search greatly outperforms greedy decoding in response refinement and surpasses other methods, such as best-of-N refinement or simple iterative refinement. Furthermore, tree search is essential for improving judgment capability, especially against adversarial inputs, as indicated in Table 5. Similar responses with opposite labels generated during the tree-search process can enhance robustness against challenging scenarios. Moreover, the results presented in Tables 4 and 5 underscore the importance of iterative training for both the actor and the refiner. This iterative training process ensures mutual improvement, which is crucial for the overall effectiveness of our framework.

**Scaling test-time compute significantly boosts model performance.** Inspired by the recent developments in test-time compute scaling (Snell et al., 2024), we investigate various decoding strategies during inference on SPAR-8B-DPO-iter3. Figure 5 shows that increasing inference times remarkably enhances model performance, outperforming the results of greedy decoding. Notably, while tree search refinement's performance growth is slower, it ultimately achieves superior results compared to best-of-N generation. This indicates that

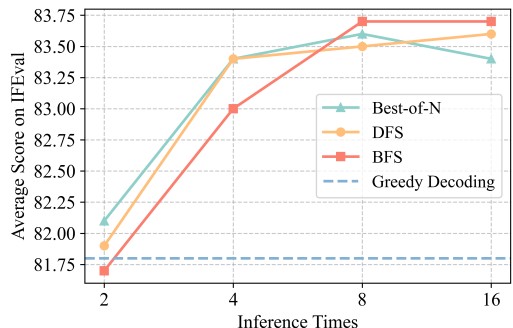

Figure 5: Comparison of decoding strategies.

refinement is more powerful than generation and could be better suited for scaling test-time compute in the instruction-following task.

## 4 RELATED WORK

### 4.1 INSTRUCTION FOLLOWING

Instruction-following is a fundamental capability of LLMs and is central to LLM alignment (Ouyang et al., 2022; Cheng et al., 2023; Lou et al., 2024). Many studies have evaluated instruction-following capabilities from various perspectives (Li et al., 2023b; Zheng et al., 2023; Zeng et al., 2023; Liu et al., 2023a; Xia et al., 2024). With the expanding application of LLMs, the tasks they are expected to perform become more intricate (Liu et al., 2023b), often involving composite instructions with numerous constraints. Consequently, several benchmarks have been developed to test LLMs' ability to follow these complex instructions (Zhou et al., 2023; Jiang et al., 2023b; Qin et al., 2024; Wen et al., 2024). Additionally, multiple studies have focused on enhancing LLMs' instruction-following capabilities (Lou et al., 2023; Zhou et al., 2024; Sun et al., 2024). One crucial aspect of the instruction-following task is that subtle differences in responses can significantly impact their correctness (Zhou et al., 2023). Considering this, we introduce SPAR framework to construct preference pairs that reduce extraneous elements to highlight these subtle variations for effective improvements.

### 4.2 AUTONOMOUS LLM ALIGNMENT

Given the high cost of manually collecting alignment data, many studies focus on exploring autonomous LLM alignment methods (Cao et al., 2024). One common strategy involves using data distilled from advanced models to improve less powerful ones (Peng et al., 2023; Xu et al., 2023; Cheng et al., 2024). Alternatively, as the LLMs become stronger, several studies (Wang et al., 2023; Yuan et al., 2024; Zhang et al., 2024) investigate how to self-evolving LLMs' capabilities. Self-Instruct (Wang et al., 2023) generates instructions by employing the model's in-context learning ability. Reinforced Self-Training (Gulcehre et al., 2023) samples data from an LLM policy and utilizes the dataset to enhance the policy through offline RL algorithms. Moreover, recent research has incorporated feedback from diverse sources. SELF (Lu et al., 2023) trains LLMs to acquire meta-skills of self-feedback and self-refinement, enabling the models to self-evolve iteratively. AutoIF (Dong et al., 2024) introduces the code execution feedback. Self-rewarding (Yuan et al., 2024) and Meta-rewarding (Wu et al., 2024) leverage the LLM-as-judge ability to evaluate its own responses, thereby constructing preference pairs. However, these methods usually direct sample multiple independent responses from the actor model, which is likely to introduce the interfering factors and thus affect the model's capture of the key differences. Thus, we propose a new framework that constructs preference pairs by self-refining the model's responses, minimizing extraneous elements, and promoting more effective autonomous improvement.

## 5 CONCLUSION

In this study, we introduce a new self-play framework, SPAR, designed to improve the instruction-following capabilities of LLMs through training with refinement pairs. We reveal that, unlike traditional approaches that rely on sampling multiple independent responses from the model to construct preference pairs, refining preference pairs to minimize extraneous factors and highlight key differences lead to significant improvements in instruction-following tasks. Remarkably, the LLaMA3-8B-Instruct model, trained iteratively using our framework, outperforms GPT-4-Turbo on IFEval. With inference time compute scaling, its performance can be further improved. Moreover, the iterative enhancement of instruction-following, judgment, and refinement abilities brought about by SPAR underscores a promising path to continuous self-improvement.

## 6 ACKNOWLEDGEMENT

This work was supported by the National Science Foundation for Distinguished Young Scholars (with No. 62125604). This work was also supported by Tsinghua University Initiative Scientific

Research Program. We would also like to thank Zhipu AI for sponsoring GPU computing and API cost consumed in this study.

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

## A  DATASET INFORMATION

**Constraint Taxonomy.**  We take the taxonomy from Cheng et al. (2024), and further refine it to be more comprehensive to ensure the diversity of our prompts. The refined taxonomy is shown in Figure 6.

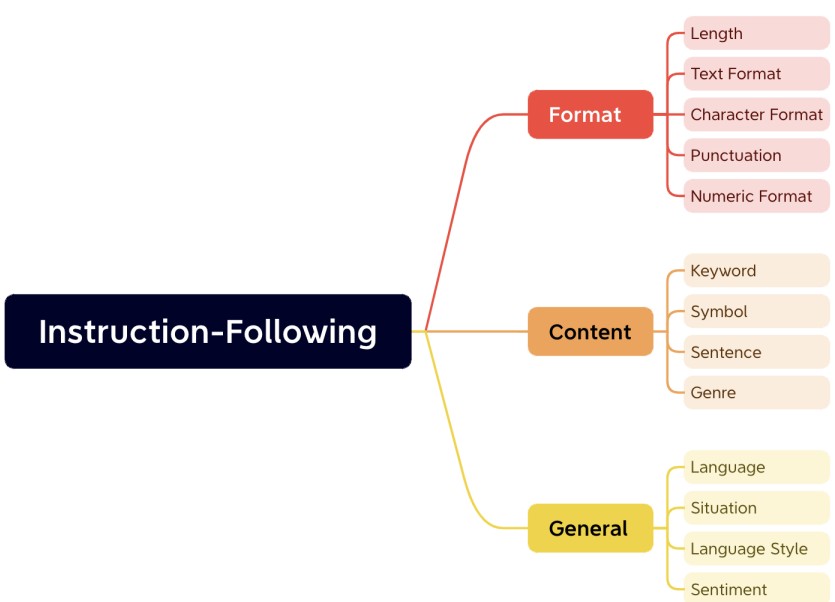

Figure 6: The detailed taxonomy of constraints for prompt evolution.

**Prompt Template.**  Here, we give the prompt for constructing complex prompts in Figure 7. For the refiner, the prompt template for judgment is provided in Figure 8. As for the refinement task, we form it as a multi-turn task after judgment, with the prompt template provided in Figure 8.

## B  TREE-SEARCH ALGORITHM

We show the detailed process of BFS and DFS refinement in Algorithm 1 and Algorithm 2.

## C  IMPLEMENTATION DETAILS

The SFT dataset for the actor comprises 8k examples, while the refiner dataset includes approximately 9k examples for judgment training and 3k for refinement training, formatted as a multi-turn

---

**Prompt Construction Template**

In this task, you need to refine the prompt to make it more specific and complex. Transfer the prompt to an instruction-following task.

To enhance specificity, refer to constraints from the following taxonomy:
{Taxonomy}

prompt: [[start]]{}[[end]]

Main constraint:
{Main Constraint}

Note:
1. You must include the main constraint (unless the main constraint contradicts the original prompt), and need to add 1~3 other constraints.
2. You need to ensure that these constraints are reasonable and do not conflict with each other.
3. The added constraints need to make sense with the original prompt and not conflict with it.
4. Please ensure the constraints as much diverse as possible, you can add new types not mentioned in the taxonomy above.
5. The priority is given to constraint types other than length constraints.
6. Do not list the added constraints in points.
7. Don't give the response to your refined prompt.
8. If the Prompt is a mathematical or coding problem, just output "None" as the refined prompt.
9. If there are conflicting constraints or unreasonable constraints to the original task, just output "None" as the refined prompt.
10. Never involve visual elements.

output in the following format:
[[start]]{your refined prompt}[[end]]

---

Figure 7: The prompt template applied for prompt evolution.

---

**Judgement Template**

Please act an expert in evaluating the capabilities of instruction-following. In the instruction-following task, the Output needs to honestly/precisely/closely follow the given Prompt.
Your task is to carefully judge whether the Output to honestly/precisely/closely follows the given Prompt. If there are any constraints in the Prompt that are not satisfied by the Output, please list all the constraints that are not satisfied.

Prompt: "{Instruction}"

Output: "{Response}"

Please carefully judge if each constraint is perfectly satisfied and give a final judgement weather the Output accurately follows the Prompt in the following format:
Step-by-step verification: xxx
Final Judgement (if the Output accurately follows the Prompt): (Yes or No)

---

**Refinement Template**

Based on your judgement, refine the Output to make sure it can honestly/precisely/closely follows the given Prompt.

Please carefully refine the Output to meet all the constraints in the Prompt.

Please format like this:
Reflection on how to refine the Output: xxx
Final Refined Output: [[start]] xxx [[end]]

---

Figure 8: The prompt template applied for the refiner's judgment and refinement.

| **Algorithm 1** BFS-Refinement | **Algorithm 2** DFS-Refinement |
|---|---|
| **Require:** Instruction $x$, Response $y$, Judgment $j$, Refiner $R_N$, depth limit $d$, branch limit $b$. 
 $\quad S_0 \leftarrow \{x, y, j\}$ 
 $\quad$ **for** $t = 1, \cdots, d$ **do** 
 $\quad\quad S'_t \leftarrow \{[x, y'] \mid s \in S_{t-1}, y' \in R_N(s, b)\}$ 
 $\quad\quad V_t \leftarrow R_N(S'_t)$ $\quad\quad\quad$ ▷ get judgment 
 $\quad\quad S_t \leftarrow \{[x, y', j'] \mid s \in S'_t, j' \in V_t(s)\}$ 
 $\quad$ **end for** 
 $\quad$ **return** $\arg\max_{s \in S_T} V_T(s)$ | **Require:** Current state $s$, depth $t$, Refiner $R_N$, depth limit $d$, threshold $v_{th}$, branch limit $b$ 
 $\quad$ **if** $t > T$ **then** record output $s = (x, y', j')$ 
 $\quad$ **end if** 
 $\quad$ **for** $s' \in R_N(s, b)$ **do** $\quad\quad$ ▷ refinement 
 $\quad\quad$ **if** $R_N(s') < v_{th}$ **then** $\quad\quad$ ▷ judgment 
 $\quad\quad\quad$ DFS$(s', t+1)$ 
 $\quad\quad$ **end if** 
 $\quad$ **end for** |

task following the first turn's judgment. These two datasets are both constructed with GPT-4o-Mini. Both the actor and refiner are trained with a learning rate of 2e-6 and a warmup ratio of 0.1, using the AdamW optimizer with $\beta_1 = 0.9$ and $\beta_2 = 0.999$. The actor is trained over 5 epochs with a batch size of 64, and the refiner is trained for 3 epochs with the same batch size. In the data construction process, we set a tree search budget of 15 to strike a balance between performance and efficiency. The average number of expanded tree nodes is around 3.7 in our experiments, which is an acceptable level. Specifically, for LLaMA3-8B-Instruct, the average expanded node numbers are 4.3, 3.7, and 3.4 across different iterations, demonstrating a decreasing trend as the model becomes stronger. For the actor iterative training, each iteration uses around 5k examples for DPO. To enhance training stability as suggested by (Hou et al., 2024), an additional SFT loss is added to the chosen response with a weight of 0.1. Here, the learning rate is set to 2e-7, $\beta$ to 0.1, with a warmup ratio of 0.1, and training is conducted for 1 epoch with a batch size of 32. For the refiner, each iteration utilizes about 10k examples, including 4k refinement samples. We ensure the judgment training dataset maintains a balance of positive and negative samples. The training configuration remains the same as for SFT, except the learning rate is set to 1e-6. All experiments are performed on an $8 \times 80$G Nvidia A100 setup.

For our baseline methods, we have maintained uniform settings to ensure fairness. For SELF, we initialize with our constructed datasets, $D_{Actor}$ and $D_{Refiner}$. In the case of self-rewarding and meta-rewarding, we start with $D_{Actor}$ and $D_{JSFT}$. For Humpback, we create the seed dataset by combining about 3k data from the Oasst[1] dataset and 5k data from $D_{Actor}$. We also control the number of training samples to be nearly identical for fair comparisons.

## D   BASELINES

We compare our method with four popular self-improvement approaches, including:

- **AutoIF** (Dong et al., 2024) incorporates code feedback and online DPO training to improve instruction-following ability in both distillation and self-evolution settings.
- **SELF** (Lu et al., 2023) proposes leveraging language feedback to guide response generation in order to achieve iterative self-improvement.
- **Self-rewarding** (Yuan et al., 2024) proposes to combine the reward model and policy model to enhance alignment capabilities simultaneously.
- **Meta-rewarding** (Wu et al., 2024) further introduces a meta-judge to address judgment capability limitations, building on the self-rewarding framework.
- **Humpback** (Li et al., 2023a) proposes training an instruction generation model to synthesize high-quality data using web resources.

## E   EXPERIMENT RESULTS

### E.1   INSTRUCTION-FOLLOWING EVALUATION RESULTS.

The evaluation results on instruction-following benchmarks are shown in Table 6. Our method outperforms all baselines on these benchmarks and show substantial improvements in each iteration (Figure 9).

---

[1] `https://huggingface.co/datasets/OpenAssistant/oasst1`

## E.2    GENERAL PERFORMANCE EVALUATION

Our analysis in Table 7 reveals that SPAR training not only doesn't harm general performance, but it can also even bring enhancements.

## E.3    JUDGMENT EVALUATION RESULTS.

As shown in Table 8, the judgment capability improves in each iteration and the accuracy outperforms all baselines.

## E.4    ABLATION STUDY ON JUDGMENT CAPABILITY.

In our experiments, we employ majority voting for iterative improvements for judgment capability. We show the results of the refiner SPAR-8B-SFT's sampling times and performance on LLMBar in Table 9. To balance the performance and computation time, we choose majority voting@5.

## E.5    ABLATION STUDY ON REFINEMENT CAPABILITY.

Table 10 shows the results of different decoding strategies for the refinement task on SPAR-8B. For methods except greedy decoding, we use the same inference budget. We can see that the tree search algorithms largely outperform other methods, verifying the importance of incorporating tree search refinement.

## E.6    INFERENCE-TIME SCALING COMPARISON

Figure 10 presents a comparison between SPAR and self-rewarding, focusing on their scalability with regard to inference times, measured by the number of response generations in our study. Our analysis includes both the LLaMA3-8B-Instruct and Mistral-7B-Instruct models. The results demonstrate that SPAR outperforms the self-rewarding method when additional computational resources are allocated for inference time, leading to enhanced performance.

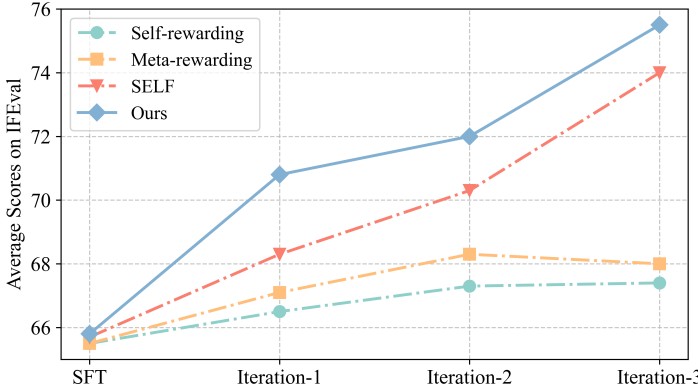

Figure 9: Comparison with baseline methods across iterations. SPAR-7B consistently surpasses all baselines.

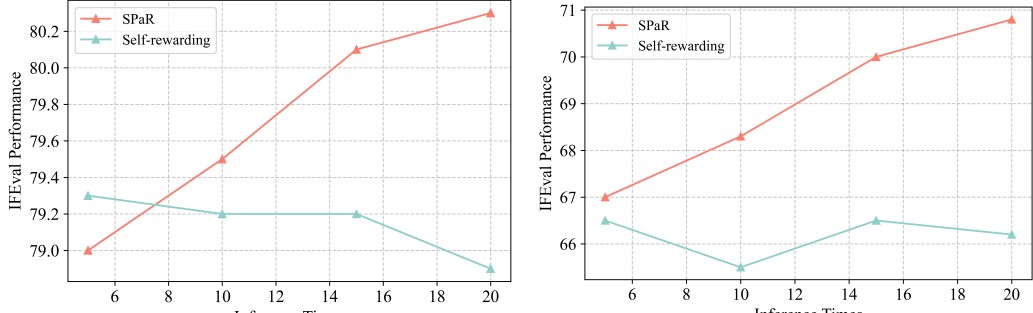

Figure 10: Inference-time scaling comparison on IFEval. The left panel showcases results for LLaMA3-8B-Instruct, while the right panel presents findings for Mistral-7B-Instruct.

Table 6: Full results of SPAR-7B, SPAR-9B, and SPAR-70B on instruction-following benchmarks. P stands for prompt level, and I represents instruction level. L and S denote loose and strict evaluations, respectively. Avg. indicates average results and Lv means level. Scores marked with [†] are sourced directly from the original paper.

| Model | IFEval | | | | | FollowBench (SSR) | | | | | |
|---|---|---|---|---|---|---|---|---|---|---|---|
| | P (L) | I (L) | P (S) | I (S) | Avg. | Lv-1 | Lv-2 | Lv-3 | Lv-4 | Lv-5 | Avg. |
| *Mistral-7B Models* | | | | | | | | | | | |
| Mistral-7B-Instruct | 55.1 | 64.9 | 49.9 | 60.2 | 57.5 | 65.1 | 61.6 | 61.6 | 56.8 | 57.2 | 60.4 |
| SELF | 71.3 | 79.7 | 68.0 | 76.9 | 74.0 | 71.5 | 64.2 | 60.8 | 58.0 | 57.0 | 62.3 |
| Humpback | 60.4 | 71.0 | 56.6 | 67.6 | 63.9 | 70.7 | 63.9 | 63.8 | 59.8 | 57.9 | 63.2 |
| Self-Rewarding | 64.3 | 73.5 | 61.0 | 70.7 | 67.4 | 70.8 | 64.8 | 62.3 | 61.9 | **58.3** | 63.6 |
| Meta-Rewarding | 65.1 | 74.7 | 61.0 | 71.1 | 68.0 | 73.2 | 64.6 | 64.5 | 60.6 | 57.6 | 64.1 |
| SPAR-7B-SFT | 62.7 | 72.3 | 59.3 | 68.7 | 65.8 | 74.4 | 64.3 | 62.5 | 58.2 | 55.0 | 62.9 |
| SPAR-7B-DPO-iter1 | 68.2 | 76.6 | 64.7 | 73.6 | 70.8 | 73.2 | 64.6 | 63.1 | 60.3 | 56.6 | 63.6 |
| SPAR-7B-DPO-iter2 | 70.0 | 78.1 | 65.8 | 74.2 | 72.0 | 72.2 | **65.7** | 61.4 | **62.4** | 57.5 | 63.8 |
| SPAR-7B-DPO-iter3 | **74.1** | **80.9** | **69.7** | **77.1** | **75.5** | **74.6** | 63.8 | **66.1** | 61.0 | 58.0 | **64.7** |
| *GLM-4-9B Models* | | | | | | | | | | | |
| GLM-4-9B-Chat | 71.5 | 79.9 | 68.0 | 77.2 | 74.2 | 80.8 | 75.1 | 67.4 | 64.3 | **65.4** | 70.6 |
| SPAR-9B-SFT | 71.5 | 80.5 | 68.8 | 78.1 | 74.7 | 79.4 | 70.9 | **68.2** | 65.1 | 63.7 | 69.5 |
| SPAR-9B-DPO-iter1 | 73.8 | 81.2 | 70.6 | 78.5 | 76.0 | 82.6 | 76.0 | 67.9 | 64.9 | 63.6 | 71.0 |
| SPAR-9B-DPO-iter2 | 76.7 | 83.3 | 73.2 | 80.9 | 78.5 | 80.4 | 76.6 | 67.4 | **68.7** | 64.1 | 71.4 |
| SPAR-9B-DPO-iter3 | **77.3** | **84.1** | **73.6** | **81.4** | **79.1** | **82.7** | **76.7** | 67.9 | 68.3 | 64.2 | **72.0** |
| *LLaMA3-70B Models* | | | | | | | | | | | |
| LLaMA3-70B-Instruct | 83.7 | 88.9 | 77.1 | 83.8 | 83.4 | 77.1 | 72.5 | 69.4 | 68.7 | 66.3 | 70.8 |
| AutoIF-70B[†] | **85.6** | **90.4** | 80.2 | 86.7 | 85.7 | 71.0 | 67.2 | 66.2 | 64.6 | 63.5 | 66.5 |
| SPAR-70B-DPO-iter1 | 84.5 | 89.2 | 80.2 | 85.7 | 84.9 | 77.6 | 74.0 | 70.2 | 70.6 | 66.9 | 71.9 |
| SPAR-70B-DPO-iter2 | 85.0 | 89.4 | 81.5 | 87.2 | 85.8 | **80.4** | **76.4** | 69.9 | **73.7** | 70.2 | 74.1 |
| SPAR-70B-DPO-iter3 | **85.6** | 90.2 | **81.3** | **87.3** | **86.1** | 80.3 | 75.7 | **71.4** | **73.7** | **70.5** | **74.3** |

Table 7: Performance on general benchmarks. SPAR maintains the model's general capabilities.

| Model | GSM8k | TriviaQA | MMLU | HumanEval | Average |
|---|---|---|---|---|---|
| *Mistral-7B Models* | | | | | |
| Mistral-7B-Instruct | 42.9 | 72.5 | 57.9 | 32.9 | 51.6 |
| SPAR-7B-SFT | 56.4 | 72.8 | 56.7 | 44.5 | 57.6 (+6.0) |
| SPAR-7B-DPO-iter1 | 55.6 | 72.2 | 55.3 | 46.3 | 57.4 (+5.8) |
| SPAR-7B-DPO-iter2 | 54.4 | 72.1 | 55.8 | 45.1 | 56.9 (+5.3) |
| SPAR-7B-DPO-iter3 | 58.2 | 71.6 | 55.1 | 46.3 | 57.8 (+6.2) |
| *LLaMA3-8B Models* | | | | | |
| LLaMA3-8B-Instruct | 75.4 | 75.9 | 63.6 | 55.5 | 67.6 |
| SPAR-8B-SFT | 75.6 | 76.0 | 64.0 | 61.6 | 69.3 (+1.7) |
| SPAR-8B-DPO-iter1 | 78.8 | 75.2 | 63.8 | 60.4 | 69.6 (+2.0) |
| SPAR-8B-DPO-iter2 | 77.0 | 74.9 | 63.1 | 60.4 | 68.9 (+1.3) |
| SPAR-8B-DPO-iter3 | 77.7 | 75.1 | 63.1 | 60.9 | 69.2 (+1.6) |
| *GLM-4-9B Models* | | | | | |
| GLM-4-9B-Chat | 80.6 | 69.7 | 71.9 | 74.3 | 74.1 |
| SPAR-9B-SFT | 82.9 | 69.4 | 71.8 | 73.8 | 74.5 (+0.4) |
| SPAR-9B-DPO-iter1 | 82.6 | 68.8 | 71.6 | 75.0 | 74.5 (+0.4) |
| SPAR-9B-DPO-iter2 | 82.8 | 68.9 | 71.8 | 73.8 | 74.3 (+0.2) |
| SPAR-9B-DPO-iter3 | 83.0 | 69.0 | 72.1 | 73.2 | 74.3 (+0.2) |
| *LLaMA3-70B Models* | | | | | |
| LLaMA3-70B-Instruct | 92.2 | 87.2 | 80.8 | 79.3 | 84.9 |
| SPAR-70B-DPO-iter1 | 92.5 | 90.4 | 81.0 | 79.3 | 85.8 (+0.9) |
| SPAR-70B-DPO-iter2 | 92.9 | 89.5 | 80.4 | 78.7 | 85.4 (+0.5) |
| SPAR-70B-DPO-iter3 | 93.4 | 86.7 | 80.6 | 79.9 | 85.2 (+0.3) |

Table 8: Judgment evalution results on LLMBar for SPAR-7B. Acc. stands for accuracy.

| Model | Natural | | Adversarial | | | | | | | | | | | | Average | |
| | | | GPTInst | | GPTOut | | Manual | | Neighbor | | Average | | | | | |
| | Acc. | F1 | Acc. | F1 | Acc. | F1 | Acc. | F1 | Acc. | F1 | Acc. | F1 | | | Acc. | F1 |
|---|---|---|---|---|---|---|---|---|---|---|---|---|---|---|---|---|
| Mistral-7B-Instruct | 58.0 | **69.1** | 57.1 | **68.8** | 50.0 | **64.1** | 45.6 | **61.5** | 47.8 | 62.6 | 50.1 | **64.3** | | | 51.7 | **65.2** |
| SELF | 68.0 | 65.2 | 71.2 | 68.7 | 56.4 | 56.8 | 62.0 | 52.6 | 67.5 | 62.3 | 64.3 | 60.1 | | | 65.0 | 61.1 |
| Self-Rewarding | 68.0 | 64.0 | 69.0 | 63.7 | 59.6 | 53.7 | **63.0** | 57.5 | **69.4** | **64.3** | **65.3** | 59.8 | | | 65.8 | 60.6 |
| Meta-Rewarding | 67.5 | 62.4 | 71.7 | 68.7 | 56.4 | 51.8 | **63.0** | 56.4 | 66.8 | 62.1 | 64.5 | 59.7 | | | 65.1 | 60.3 |
| SPAR-7B-SFT | 69.5 | 63.9 | 71.7 | 67.5 | 55.3 | 48.8 | 55.4 | 45.3 | **69.4** | 62.3 | 63.0 | 56.1 | | | 64.3 | 57.6 |
| SPAR-7B-RFT-iter1 | 67.0 | 62.1 | 66.3 | 62.7 | 56.4 | 52.9 | 60.9 | 52.6 | 64.2 | 60.7 | 61.9 | 57.2 | | | 63.0 | 58.2 |
| SPAR-7B-RFT-iter2 | 68.0 | 64.4 | 68.5 | 64.6 | **60.6** | 57.5 | 62.0 | 52.1 | 64.2 | 60.0 | 63.8 | 58.5 | | | 64.7 | 59.7 |
| SPAR-7B-RFT-iter3 | **71.0** | 66.7 | **72.3** | 67.5 | 57.4 | 55.6 | 60.9 | 51.4 | 68.3 | 62.6 | 64.7 | 59.2 | | | **66.0** | 60.7 |

Table 9: Comparison of decoding strategies on LLMBar.

| Method | Natural | | Adversarial | |
| | Acc. | F1 | Acc. | F1 |
|---|---|---|---|---|
| Greedy Decoding | 68.0 | 60.7 | 63.9 | 55.1 |
| Majority Voting@3 | 69.0 | 60.8 | 63.7 | 54.5 |
| Majority Voting@5 | 68.5 | 60.9 | 64.7 | 56.5 |
| Majority Voting@7 | 66.5 | 58.8 | 65.7 | 56.7 |
| Majority Voting@9 | 69.0 | 61.2 | 65.8 | 57.1 |

Table 10: Comparison of different decoding strategies for refinement task. Acc-GPT stands for the accuracy of using GPT-4o as judge, and Acc-SPAR for the accuracy of using SPAR-8B-RFT-iter3 as judge.

| Method | Acc-GPT | Acc-SPAR |
|---|---|---|
| Greedy Decoding | 69.5 | 65.0 |
| Best of N | 74.0 | 80.0 |
| Iterative Refinement | 71.0 | 82.0 |
| BFS | **79.0** | **90.5** |
| DFS | **79.0** | 90.0 |

