# OpenReview forum: "SPaR: Self-Play with Tree-Search Refinement to Improve Instruction-Following in Large Language Models"
_ICLR.cc/2025/Conference — ICLR 2025 Poster_

### Official Review · Reviewer_AshF · 2024-10-17

**Soundness:** 3
**Presentation:** 4
**Contribution:** 3
**Rating:** 8
**Confidence:** 4

**Summary:**

The authors introduce SPAR, an automated and scalable approach designed for self-improvement in instruction-following tasks through self-play. The core idea is to create paired responses with minimal irrelevant variations, allowing for precise training of the model's instruction-following capabilities. In the SPAR framework, the authors fully leverage test-time scaling: using tree search to obtain higher-quality data for training the model's instruction-following abilities, and using self-consistency to acquire higher-quality data for training the model's discriminative and refinement abilities. Experimental results show that the SPAR framework significantly outperforms various self-critique baselines.

**Strengths:**

- Constructing tailored and distinct instruction-following response pairs for the model by eliminating irrelevant content is a strong motivation for enhancing the model's instruction-following abilities.
- The SPAR framework's proposal to use test-time scaling during the training phase to obtain high-quality data for training the model's
- The experimental setup is reasonable, and the results appear promising.
-  The writing in the paper is clear and easy to understand.

**Weaknesses:**

Using test-time scaling (more accurately, inference-time scaling) during the training phase to obtain high-quality data for self-critique is well-motivated, but it undoubtedly introduces significant training overhead. Therefore, providing a detailed comparison of the training costs of different methods, or comparing the gains when the costs are aligned, would make the paper's conclusions more convincing.

typo: line 239 needs a blank after 'refiner'.

**Questions:**

See weakness.

---

> ### Author Response · Authors · 2024-11-19
> **Response to Reviewer AshF**
>
> Dear Reviewer AshF,
>
> We are deeply thankful for your positive feedback and for acknowledging the novelty and significance of our contributions. Your recognition of our method's scalability, the motivation behind leveraging test-time scaling for training, the promising experimental results, and the clarity of our writing are genuinely encouraging. We are especially honored by your clear and strong support!
>
>
> **1. On Higher Costs**
> > Weakness 1: Inference-time scaling is well-motivated for model improvement but will introduce higher costs.
>
> Thank you for your insightful comment! As the ultimate goal is to enhance the model's performance in instruction following, inference-time scaling is actually more efficient than traditional pretraining solutions or creating high-quality data with human annotators.
>
> We have calculated the average number of expanded tree nodes in our framework and included these details in Appendix C. Specifically, with the LLaMA3-8B model, the average number of expanded tree nodes is around 3.8, which is within an acceptable range of added computation.
>
> Moreover, to effectively understand the costs and gains, we have conducted an additional series of experiments controlling the maximum number of response generations (inference times) during the training data construction process. We perform these experiments on two models, LLaMA3-8B and Mistral-7B, and report the average performance on IFEval. Here are the results:
>
> | Model     | Method | 5    | 10   | 15   | 20   |
> |---|-|--|--|--|--|
> | LLaMA3-8B    | Self-rewarding | 79.3 | 79.2 | 79.2 | 78.9 |
> | | SPaR | 79.0 | 79.5 | 80.1 | 80.3 |
> | Mistral-7B   | Self-rewarding | 66.5 | 65.5 | 66.5 | 66.2 |
> | | SPaR | 67.0 | 68.3 | 70.0 | 70.8 |
>
> The results indicate that SPaR surpasses the Self-rewarding method when additional computational resources are allocated for inference. This highlights the advantage of our method in scaling inference-time costs to achieve superior performance.
>
> **2. Clarification of Writing Issue**
> > Weakness 2: Typo in line 239
>
> Thank you for your careful and detailed reading! We have corrected this issue and thoroughly reviewed the entire paper to address any other errors.

---

> > ### Comment · Reviewer_AshF · 2024-11-20
> >
> > The direct comparison with Self-rewarding under controlled cost conditions has enhanced my confidence in this work, which is suggested to be added in the revised version. Thank the author for addressing my concerns and I will maintain my rating.

---

### Official Review · Reviewer_7aGY · 2024-11-01

**Soundness:** 3
**Presentation:** 3
**Contribution:** 3
**Rating:** 6
**Confidence:** 4

**Summary:**

This paper proposes a novel self-play framework (called SPAR) integrating tree-search self-refinement to yield valid and comparable preference pairs free from distractions, so as to teach models to recognize the key differences that lead to improved instruction following. To gain a good start for Actor and Refiner, the authors construct a high-quality dataset with 43K complex instruction-following prompts and an SFT dataset for improving the instruction-following capabilities of LLMs. Through extensive experiments on several LLMs, the authors demonstrate the effectiveness of proposed SPAR.

**Strengths:**

* The proposed approach is intuitive and has strong motivation.

* This paper is well-written and presents clear ideas.

* The authors conduct extensive experiments to validate the effectiveness of proposed SPAR.

**Weaknesses:**

* **SPAR introduces additianal training overheads.** SPAR initially requires constructing additional data to train the actor and trainer. Building on this, it needs to incorporate iterative training with tree search and self-consistency, which greatly increases the training cost compared to self-rewarding.

* **Some crucial information is missing in the experiment section.**  For example, what is the average number of search nodes in the tree search, and does it decrease with the iterations? How does LLaMA3-70B perform at different iterations (SPAR-70B-SFT, SPAR-70B-DPO-iter1, SPAR-70B-DPO-iter2)?

**Questions:**

See weaknesses.

In addition:

(1) line 527, GPT-4-Turbo or GPT-4o-mini?

(2) Can you compare the training cost of SPAR, Self-Rewarding and Meta-Rewarding?

(3) Does more iteration brings higher performance?

---

> ### Author Response · Authors · 2024-11-19
> **Response to Reviewer 7aGY**
>
> Dear Reviewer 7aGY,
>
> Thanks for your comprehensive and detailed suggestions for our work! We really value your comments on the training overheads and clarification of experimental details. We hope our responses below could address your concerns:
>
> **1. On Training Overheads**
> > Weakness 1: SPaR introduces additianal training overheads.
>
> Thank you for your insightful comment! As the ultimate goal is to enhance the model's performance in instruction following, inference-time scaling is actually more efficient than traditional pretraining solutions or creating high-quality data with human annotators.
>
> We have calculated the average number of expanded tree nodes in our framework and included these details in Appendix C. Specifically, with the LLaMA3-8B model, the average number of expanded tree nodes is around 3.8, which is within an acceptable range of added computation.
>
> Moreover, to effectively understand the costs and gains, we have conducted an additional series of experiments controlling the maximum number of response generations (inference times) during the training data construction process. We perform these experiments on two models, LLaMA3-8B and Mistral-7B, and report the average performance on IFEval. Here are the results:
>
> | Model     | Method | 5    | 10   | 15   | 20   |
> |---|-|--|--|--|--|
> | LLaMA3-8B    | Self-rewarding | 79.3 | 79.2 | 79.2 | 78.9 |
> | | SPaR | 79.0 | 79.5 | 80.1 | 80.3 |
> | Mistral-7B   | Self-rewarding | 66.5 | 65.5 | 66.5 | 66.2 |
> | | SPaR | 67.0 | 68.3 | 70.0 | 70.8 |
>
> The results indicate that SPaR surpasses the Self-rewarding method when additional computational resources are allocated for inference. This highlights the advantage of our method in scaling inference-time costs to achieve superior performance.
>
> **2. Clarification of Experimental Details**
> > Weakness 2: Some experimental details are not shown in paper.
>
> Thank you for your valuable suggestions! We have included the average number of search nodes in our experiments in Appendix C. For LLaMA3-8B, the number of average expanded nodes is 4.3, 3.7, and 3.4 across different iterations, demonstrating a decreasing trend as the model becomes better. The performance of LLaMA3-70B at each iteration, previously omitted due to space constraints, has now been added to Appendix D.1 Table 9.
>
> > Question 1: line 527, GPT-4-Turbo or GPT-4o-mini?
>
> The model is GPT-4-Turbo. SPaR-trained LLaMA3-8B-Instruct outperforms the GPT-4-Turbo on the IFEval benchmark. The performance of GPT-4-Turbo is derived from the original benchmark [1].
>
> [1] Zhou, Jeffrey, et al. "Instruction-following evaluation for large language models." arXiv preprint arXiv:2311.07911 (2023).
>
>
> > Question2: Can you compare the training cost of SPAR, Self-Rewarding and Meta-Rewarding?
>
> We would like to clarify that the training costs for all three methods are nearly identical, as we have controlled the number of training samples to ensure fairness.
>
> The inference times required for data construction vary among these methods. For instance, in the case of LLaMA3-8B, the average number of responses generated by Self-Rewarding and Meta-Rewarding methods is 5, whereas for SPaR, it is approximately 8.8. This increase is within an acceptable range. As mentioned in our response to Weakness 1, we have conducted experiments to compare the costs and gains.
>
>
> **3. On Iterative Improvement**
> > Question3: Does more iteration bring higher performance?
>
> Thank you for your insightful comment! Additional iterations can generally improve the model's performance but with diminishing returns.
> For instance, in our experiments with LLaMA3-8B, extending to a fourth iteration showed smaller improvements of an average of 0.3.
> This can be caused by model capacity limitations or challenges in iterative DPO training.
>
> Furthermore, most self-training methods, such as Self-rewarding and SELF, typically use three iterations. We follow this practice to make it easier to compare our results with these methods.

---

> > ### Comment · Reviewer_7aGY · 2024-11-22
> >
> > Thanks for the detailed response. Currently, most of my concerns are resolved and I maintain my score.

---

### Official Review · Reviewer_2gma · 2024-11-04

**Soundness:** 2
**Presentation:** 2
**Contribution:** 3
**Rating:** 6
**Confidence:** 4

**Summary:**

The paper introduces SPAR (Self-Play with Tree-Search Refinement), a self-improvement framework that enhances the instruction-following capabilities of LLMs by minimizing extraneous factors and highlighting key differences in preference pairs. This method involves an iterative training process where a model (actor) performs tasks, and a paired model (refiner) evaluates and refines the imperfect responses using a tree-search algorithm through structured feedback loops. The authors evaluate SPAR with two LLMs on the IFEval and FollowBench benchmarks. Additionally, they contribute a dataset with 43k complex instruction-following prompts and an SFT dataset that can improve the instruction-following capabilities of LLMs.

**Strengths:**

1. Effective in reducing noise. By minimizing content variations in preference pairs, SPAR helps the model focus on essential elements, which improves its instruction-following accuracy.

2. Comprehensive ablation experiments. The authors conducted extensive ablation studies to verify the impact of interfering factors on preference learning and to assess the rationality of each component in the framework.

3. Generalization without degradation. The approach does not degrade general language model capabilities, suggesting a balanced enhancement in alignment without compromising overall functionality.

4. Contribution of datasets. The authors provide valuable datasets that benefit the development of this research area.

**Weaknesses:**

1. Limited validation across models. The effectiveness of the method was validated on only three models. Further exploration is needed to assess the framework's applicability to other models.

2. Reliance on complex setup and compute resources. The framework's iterative training, including tree-search refinement and multiple model roles, may require significant computational resources. Therefore, the performance-cost trade-off needs further clarification.

3. Lack of comparative details. The paper lacks sufficient details in its comparisons with other methods, such as how each baseline initializes the model.

**Questions:**

1. The paper only lists results from the first three iterations, and the data indicate that the model's performance still has room for improvement. Could you provide a simple analysis of when the model might reach optimal performance?

2. Does the framework heavily depend on the model's initial performance? Can it be directly applied to the raw models provided officially?

3. It is suggested to directly specify the strong LLM used in Section 2.2.2

---

> ### Author Response · Authors · 2024-11-19
> **Response to Reviewer 2gma**
>
> Dear Reviewer 2gma,
>
> Thanks for your comprehensive and detailed suggestions for our work! We really value your comment on experiments with additional models, compute resources and baseline details. We hope our detailed response could address your concerns:
>
> **1. Experiments with Additional Models**
> > Weakness 1: Limited validation across models. The effectiveness of the method was validated on only three models.
>
> Thank you for your valuable suggestion! We have added additional experiments using GLM-4-9b, a popularly used open-source LLM in the community. The results are shown below (Cf. Table 9 for full results):
>
> | Method    | IFEval | |  |  | | Follow|Bench |  |  | |  |
> |-|-|--|-|-|-|-|-|-|-|-|-|
> |    | P (L) | I (L) | P (S) | I (S) | Avg. | Lv-1 | Lv-2 | Lv-3 | Lv-4 | Lv-5 | Avg. |
> | GLM-4-9B   | 71.5     | 79.9| 68.0       | 77.2| 74.15 | 80.8    | 75.1    | 67.4    | 64.3    | 65.4    | 70.6 |
> | SPaR-9B      | 77.3     | 84.1| 73.6     | 81.4| 79.1  | 82.7    | 76.7    | 67.9    | 68.3    | 64.2    | 72.0 |
>
> SPaR significantly improves the instruction-following capabilities of GLM-4-9B. We believe this further validates the effectiveness of our method.
>
>
> **2. On Computational Resources**
> > The framework's iterative training, including tree-search refinement and multiple model roles, may require significant computational resources.
>
> Thank you for your insightful comment! As the ultimate goal is to enhance the model's performance in instruction following, inference-time scaling is actually more efficient than traditional pretraining solutions or creating high-quality data with human annotators.
>
> We have calculated the average number of expanded tree nodes in our framework and included these details in Appendix C. Specifically, with the LLaMA3-8B model, the average number of expanded tree nodes is around 3.8, which is within an acceptable range of added computation.
>
> Moreover, to effectively understand the costs and gains, we have conducted an additional series of experiments controlling the maximum number of response generations (inference times) during the training data construction process. We perform these experiments on two models, LLaMA3-8B and Mistral-7B, and report the average performance on IFEval. Here are the results:
>
> | Model     | Method | 5    | 10   | 15   | 20   |
> |---|-|--|--|--|--|
> | LLaMA3-8B    | Self-rewarding | 79.3 | 79.2 | 79.2 | 78.9 |
> | | SPaR | 79.0 | 79.5 | 80.1 | 80.3 |
> | Mistral-7B   | Self-rewarding | 66.5 | 65.5 | 66.5 | 66.2 |
> | | SPaR | 67.0 | 68.3 | 70.0 | 70.8 |
>
> The results indicate that SPaR surpasses the Self-rewarding method when additional computational resources are allocated for inference. This highlights the advantage of our method in scaling inference-time costs to achieve superior performance.
>
> **3. Clarification of Comparative Details**
> > Weakness 3: Lack of comparative details. The paper lacks sufficient details in its comparisons with other methods, such as how each baseline initializes the model.
>
> Thank you for your valuable suggestion! We have made every effort to ensure that the comparisons between baselines are fair, including the model initialization. We have expanded detailed information in Appendix C.
>
>
> **4. On Iterative Improvement**
> > Question 1: The paper only lists results from the first three iterations, and the data indicate that the model's performance still has room for improvement.
>
> Thank you for your insightful comment! Additional iterations can generally improve the model's performance but with diminishing returns.
> For instance, in our experiments with LLaMA3-8B, extending to a fourth iteration showed smaller improvements of an average of 0.3.
> This can be caused by model capacity limitations or challenges in iterative DPO training.
>
> Furthermore, most self-training methods, such as Self-rewarding and SELF, typically use three iterations. We follow this practice to make it easier to compare our results with these methods.
>
>
> **5. On Base Model Capability**
> > Question 2: Does the framework heavily depend on the model's initial performance? Can it be directly applied to the raw models provided officially?
>
> Our method can be applied directly to capable open-source models like LLaMA3-70-Instruct, as shown in the experiments.
>
> Moreover, for weaker models, like Mistral-7B-Instruct, we can use a small-sized curated dataset to bootstrap their capabilities, after which iterative training can be effectively applied.
>
>
> **6. Clarification of Experimental Details**
> > Question 3: It is suggested to directly specify the strong LLM used in Section 2.2.2
>
> Thank you for your suggestion! We have mentioned this in Appendix C Implementation Details.

---

### Official Review · Reviewer_pYV7 · 2024-11-04

**Soundness:** 4
**Presentation:** 4
**Contribution:** 3
**Rating:** 6
**Confidence:** 3

**Summary:**

This study presents SPAR, a self-play framework that enhances LLM‘s instruction-following capabilities by training with refined preference pairs. Unlike traditional methods that rely on independent response sampling, SPAR refines pairs to reduce irrelevant factors, thereby emphasizing critical distinctions, leading to notable improvements in instruction adherence. SPAR’s iterative process enhances instruction-following, judgment, and refinement, offering a pathway for continuous model improvement.

**Strengths:**

1. The motivation makes sense. The fine-grained refinement is essential for further improving the model's instruction-following abilities.
2. The design of the proposed method is sound, allowing it to effectively achieve its intended motivation.
3. The experiments are comprehensive, demonstrating relatively strong performance.

**Weaknesses:**

1. The applicability of the method may be limited. It might be suitable primarily for further improvement of models that already possess strong instruction-following capabilities, as the experiments were conducted on models that had already undergone instruction fine-tuning. Additionally, a strong LLM is required for warm-up training before iteration (This also raises concerns about the fairness of comparisons.), and one of the goals of dataset construction is to introduce more complex instructions.
2. Missing comparison with a key baseline：Self-Alignment with Instruction Backtranslation.

**Questions:**

1. Why are judgment and refinement performed by the same model? What would happen if they were separated, or combined with the actor model, using a single model for all tasks?
2. I haven't closely checked the details of the baselines. Do they use a strong LLM, or do they rely solely on the model being evolved?

---

> ### Author Response · Authors · 2024-11-19
> **Response to Reviewer pYV7**
>
> Dear Reviewer pYV7,
>
> Thank you for your thoughtful and constructive feedback and comments! We deeply appreciate your suggestions and spare no effort during this response stage to make improvements accordingly. Below are our responses:
>
> **1. On Model Capability for Self-improvement**
> > Weakness 1: The applicability of the method may be limited. It might be suitable primarily for further improvement of models that already possess strong instruction-following capabilities.
>
> Thank you for your insightful comment! Self-improvement methods are typically designed to enhance models that already possess reasonable capabilities.
> Competitive baselines, including Meta-rewarding and AutoIF, rely on strong base models. Meta-rewarding uses LLaMA3-8B-Instruct, while AutoIF uses LLaMA3-70B-Instruct.
>
> However, our method also demonstrates its applicability to less powerful models, such as Mistral-7B-Instruct, as shown in Table 9. For these models, we provide a dataset to bootstrap their capabilities before our iterative training. To ensure fair comparisons, we also used our constructed dataset to initialize the baseline methods.
>
>
> **2. New Bseline**
> > Weakness 2: Missing comparison with a key baseline: Self-Alignment with Instruction Backtranslation.
>
> Thank you for your valuable suggestion! Initially, the lack of this method's official implementation and dataset deterred us from including it. We have since reproduced this method based on the original paper and available unofficial resources (https://github.com/Spico197/Humback). Here are the results:
>
>
> | Model   | Method    | IFEval | |  |  | | Follow|Bench |  |  | |  |
> |-|-|-|--|-|-|-|-|-|-|-|-|-|
> |    |     | P (L) | I (L) | P (S) | I (S) | Avg. | Lv-1 | Lv-2 | Lv-3 | Lv-4 | Lv-5 | Avg. |
> | LLaMA3-8B  | Humpback  | 72.5 | 80.2 | 70.1 | 78.1 | 75.2 | 66.8| 66.1| 67.2| 60.2| 62.6| 64.6|
> |   | SPaR| 79.9 | 85.4 | 78.0 | 83.7 | 81.8 | 73.0  | 72.3| 70.0| 64.1| 64.7| 68.8|
> | Mistral-7B | Humpback  | 60.4 | 71.0   | 56.6 | 67.6 | 63.9 | 70.7| 63.9| 63.8| 59.8| 57.9| 63.2|
> |   | SPaR| 74.1 | 80.9 | 69.7 | 77.1 | 75.5 | 74.6| 63.8| 66.1| 61.0  | 58.0  | 64.7 |
>
> Our method has demonstrated stronger performance compared to this baseline.
>
> **3. On Task Combination**
> > Question 1: Why are judgment and refinement performed by the same model? What would happen if they were separated, or combined with the actor model, using a single model for all tasks?
>
> We have conducted additional experiments using LLaMA3-8B to explore the impact of separating judgment and refinement tasks.
>
> | Method    | IFEval | |  |  | | Follow|Bench |  |  | |  |
> |-|-|--|-|-|-|-|-|-|-|-|-|
> |    | P (L) | I (L) | P (S) | I (S) | Avg. | Lv-1 | Lv-2 | Lv-3 | Lv-4 | Lv-5 | Avg. |
> | Separate  | 77.8     | 84.2| 75.4     | 82.5| 80.0 | 73.7    | 67.0    | 67.7    | 62.7    | 64.6    | 67.1 |
> | Combined  | 78.0     | 84.7| 75.8     | 82.6| 80.3 | 75.3    | 67.7    | 67.6    | 64.7    | 62.3    | 67.5 |
>
> The results showed that the performance of combining these tasks sightly outperforms that of separating them. Additionally, since judgment and refinement naturally constitute a two-turn task, it is efficient to handle them using a single model.
>
> Our preliminary experiments indicate that combining all tasks into a single model slightly reduces performance on instruction-following benchmarks (also observed in Figure 3, as mentioned in line 377), likely due to the judgment and refinement tasks differing notably from the instruction-following task.
>
> **4. Clarification of Baseline Details**
> > Question 2: I haven't closely checked the details of the baselines. Do they use a strong LLM, or do they rely solely on the model being evolved?
>
> All baselines used in our experiments require bootstrapping or seed datasets, which can be generated either through a strong LLM or curated by human experts. We have maintained uniform settings across all baselines and our method to ensure a fair comparison.

---

> > ### Comment · Reviewer_pYV7 · 2024-11-25
> >
> > My concerns have been addressed, so I have increased my score.

---

### Author Response · Authors · 2024-11-19
**General Response**

We sincerely thank all the reviewers for their thoughtful comments and constructive suggestions, which significantly helped us strengthen our paper.
We are encouraged to see that the reviewers recognize the novelty and significance of our proposed SPAR framework (Reviewer pYV7, 2gma, 7aGY, AshF), its sound design and comprehensive experimental validation (Reviewer pYV7, 2gma, 7aGY, AshF), and the clarity of our presentation (Reviewer pYV7, 7aGY, AshF).

In response to the reviewers' feedback, we have submitted an updated version of our paper, which now includes more experimental details, the Humpback baseline comparison, additional experiments with more models such as GLM-4-9B, and a more detailed analysis of the cost associated with our method.

Several reviewers suggested including more information regarding the cost of our method. We would like to emphasize that leveraging inference-time scaling for training to improve model performance is well-motivated (as recognized by Reviewer AshF) and is actually more efficient compared to traditional pretraining solutions or producing high-quality data with human annotators.
Furthermore, our overall cost—averaging around 3.7 expanded tree nodes—is within a reasonable range, ensuring the feasibility and scalability of the approach.

---

### Meta-Review · Area_Chair_ocwR · 2024-12-24

**Metareview:**

## Summary:
The paper introduces SPaR, a self-play framework that enhances instruction-following capabilities in language models by refining preference pairs through tree-search self-refinement. Unlike traditional methods that rely on independent response sampling, SPaR minimizes irrelevant factors and emphasizes critical distinctions, leading to improved instruction adherence. Through an iterative training process, SPaR guides LLMs to recognize key differences crucial for enhanced instruction. Experimentation demonstrates the effectiveness of SPaR, with an LLaMA3-8B model trained using SPaR surpassing GPT-4-Turbo on the IFEval benchmark without compromising general capabilities. Additionally, SPaR shows promising scalability and greatly improves the performance of LLaMA3-70B. The framework sheds novel insights into continuous model improvement.

## Strengths:
1. The paper's motivation for fine-grained refinement of responses for instruction following is well-founded and crucial for continuous model improvement.
1. The tree search-based negative refinement is effective by minimizing content variations, allowing the model to focus on essential elements for enhancing instruction-following accuracy.
1. Comprehensive ablation experiments validate the impact of interfering factors on preference learning and the rationality of each component in the framework.
1. The approach maintains generalization without compromising overall language model capabilities, indicating a balanced enhancement.
1. The paper presents comprehensive experiments with clear writing.

## Weaknesses:
1. Lack of some experiments: the initial draft does apply the proposed method to finetuning weaker base models or other different models; Comparison to more baselines are needed; Using the same vs. separate models for judgment and refinement tasks; Training for more iterations. The rebuttal addressed these concerns by reporting the corresponding experimental results.
1. Concerns about the extra overhead introduced by the data exploration process (tree search and decoding). An analysis and measurement of the cost is needed. A comparison with baselines under the same computation budget is necessary. It would be better to compare the trade-off between data exploration cost and the instruction-following improvement.
1. BFS and DFS as two basic tree search algorithms have been studied. It would be more interesting to try other search algorithms such as greedy search, A*, and MCTS, and compare their performance.
1. There are several concurrent works adopting the LLM + tree search idea to generate better data for LLM finetuning. It would be helpful to include discussions with them in the related work section.

## Decision:
The reviewers raised several concerns mainly regarding experimental comparisons and computational time overhead. The authors provided further clarifications and additional experimental results in the rebuttal, which addressed most concerns, as confirmed by three out of the four reviewers. The efforts of the authors convinced the reviewers, resulting in all positive ratings (8666). The meta-reviewer is familiar with the field and carefully reads the paper, all the comments, the rebuttals, and the discussions. The paper addresses an important open challenge for the self-improvement of LLMs: how to generate more informative and relevant preference pairs. Using tree search to refine the negative responses is an effective and intuitive idea, as demonstrated by the comprehensive experiments in the paper. By strengthening the paper with all the results presented in the discussion, the meta-reviewer believes this paper is valuable and inspirational to the community, hence acceptance is recommended.

**Additional Comments On Reviewer Discussion:**

The reviewers raised several concerns mainly regarding experimental comparisons and computational time overhead. The authors provided further clarifications and additional experimental results in the rebuttal, which addressed most concerns, as confirmed by three out of the four reviewers. The efforts of the authors convinced the reviewers, resulting in all positive ratings (8666). The meta-reviewer is familiar with the field and carefully reads the paper, all the comments, the rebuttals, and the discussions.

---

### Decision · Program_Chairs · 2025-01-22

Accept (Poster)